# Plasticity in salt bridge allows fusion-competent ubiquitylation of mitofusins and Cdc48 recognition

Vincent Anton[1] , Ira Buntenbroich[1], Ramona Schuster[1], Felix Babatz[2], Tânia Simões[1], Selver Altin[1], Gaetano Calabrese[3] , Jan Riemer[3], Astrid Schauss[2], Mafalda Escobar-Henriques[1]

**Mitofusins are dynamin-related GTPases that drive mitochondrial fusion by sequential events of oligomerization and GTP hydrolysis, followed by their ubiquitylation. Here, we show that fusion requires a trilateral salt bridge at a hinge point of the yeast mitofusin Fzo1, alternatingly forming before and after GTP hydrolysis. Mutations causative of Charcot–Marie–Tooth disease massively map to this hinge point site, underlining the disease relevance of the trilateral salt bridge. A triple charge swap rescues the activity of Fzo1, emphasizing the close coordination of the hinge residues with GTP hydrolysis. Subsequently, ubiquitylation of Fzo1 allows the AAA-ATPase ubiquitin-chaperone Cdc48 to resolve Fzo1 clusters, releasing the dynamin for the next fusion round. Furthermore, cross-complementation within the oligomer unexpectedly revealed ubiquitylated but fusion-incompetent Fzo1 intermediates. However, Cdc48 did not affect the ubiquitylated but fusion-incompetent variants, indicating that Fzo1 ubiquitylation is only controlled after membrane merging. Together, we present an integrated model on how mitochondrial outer membranes fuse, a critical process for their respiratory function but also putatively relevant for therapeutic interventions.**

## Introduction

Mitochondria, central organelles in all eukaryotic kingdoms, are dynamic and constantly remodeled by fusion and fission events, allowing adaptations to metabolic conditions (Labbe et al, 2014; Pernas & Scorrano, 2016; Wai & Langer, 2016; Cao et al, 2017; Tilokani et al, 2018). Whereas most membrane fusion processes rely on SNARE proteins, mitochondrial fusion depends on large dynamin-like GTPases (Gasper et al, 2009; Han et al, 2017). They undergo self-oligomerization and drive membrane remodeling via conformational changes, stimulated by GTP hydrolysis (Daumke & Praefcke,

2018). Mitochondrial dynamin-like GTPases include the mitofusins, MFN1/2 in mammals and Fzo1 in yeast, mediating fusion between two outer membranes (OMs) (Escobar-Henriques & Anton, 2013; Kraus & Ryan, 2017). Deficiencies in MFN2 are causative of the type 2 subset of Charcot–Marie–Tooth (CMT2A) neuropathy (Zuchner et al, 2004; Barbullushi et al, 2019). The emerging diversity of CMT2A disease mutations pinpoints the complexity of the role of mitofusin (Engelhart & Hoppins, 2019; Sloat et al, 2019). Moreover, MFN2 was linked to Parkinson's disease and to disorders caused by energy-expenditure deregulation, such as cancer, obesity, and diabetes (Stuppia et al, 2015; Schrepfer & Scorrano, 2016; Cao et al, 2017; Dorn, 2019). However, despite the importance of mitochondrial fusion, the molecular details of how mitofusins drive membrane merging are remarkably unknown (Daumke & Roux, 2017).

Mitofusins are anchored to the OM by one or two transmembrane (TM) regions, flanked by a large N-terminal and a small C-terminal domain (Rapaport et al, 1998; Rojo et al, 2002; Mattie et al, 2018) (Fig 1A). The structure of the bacterial homologue of mitofusin, bacterial dynamin-like protein (BDLP), predicted that N- and C-terminal domains intertwine in the cytosol forming two helix bundles (HBs), named neck (HB1) and trunk (HB2), followed by the globular GTPase domain (Low & Lowe, 2006; Low et al, 2009). These predictions allowed obtaining crystal structures of a truncated version of human MFN1, named minimal GTPase domain (MGD). It corresponds to the GTPase and adjacent neck domain (Qi et al, 2016; Cao et al, 2017). Both full-length and MGD structure models of MFN1 predict stabilization of the HBs by amphipathic interactions, also proposed to directly contribute to membrane merging (De Vecchis et al, 2017; Daste et al, 2018; Brandner et al, 2019). Different conformations of BDLP and MFN1-MGD revealed important information on hinge points and interface residues required for dimer formation. Indeed, mitochondrial fusion requires conformational plasticity of mitofusins (Franco et al, 2016; Qi et al, 2016; Cao et al, 2017; Rocha et al, 2018; Yan et al, 2018). GTPase–GTPase (G–G) interactions allow dimerization and were proposed to mediate *trans*-tethering of mitochondria (Qi et al, 2016; Cao et al, 2017; Yan et al, 2018). In contrast, an alternative model for *trans*-interaction implied the formation of antiparallel coiled-coil

[1]Institute for Genetics, Cologne Excellence Cluster on Cellular Stress Responses in Aging-Associated Diseases (CECAD), Center for Molecular Medicine Cologne, University of Cologne, Cologne, Germany   [2]CECAD, University of Cologne, Cologne, Germany   [3]Institute for Biochemistry, Department of Chemistry, University of Cologne, Cologne, Germany

Correspondence: Mafalda.Escobar@uni-koeln.de

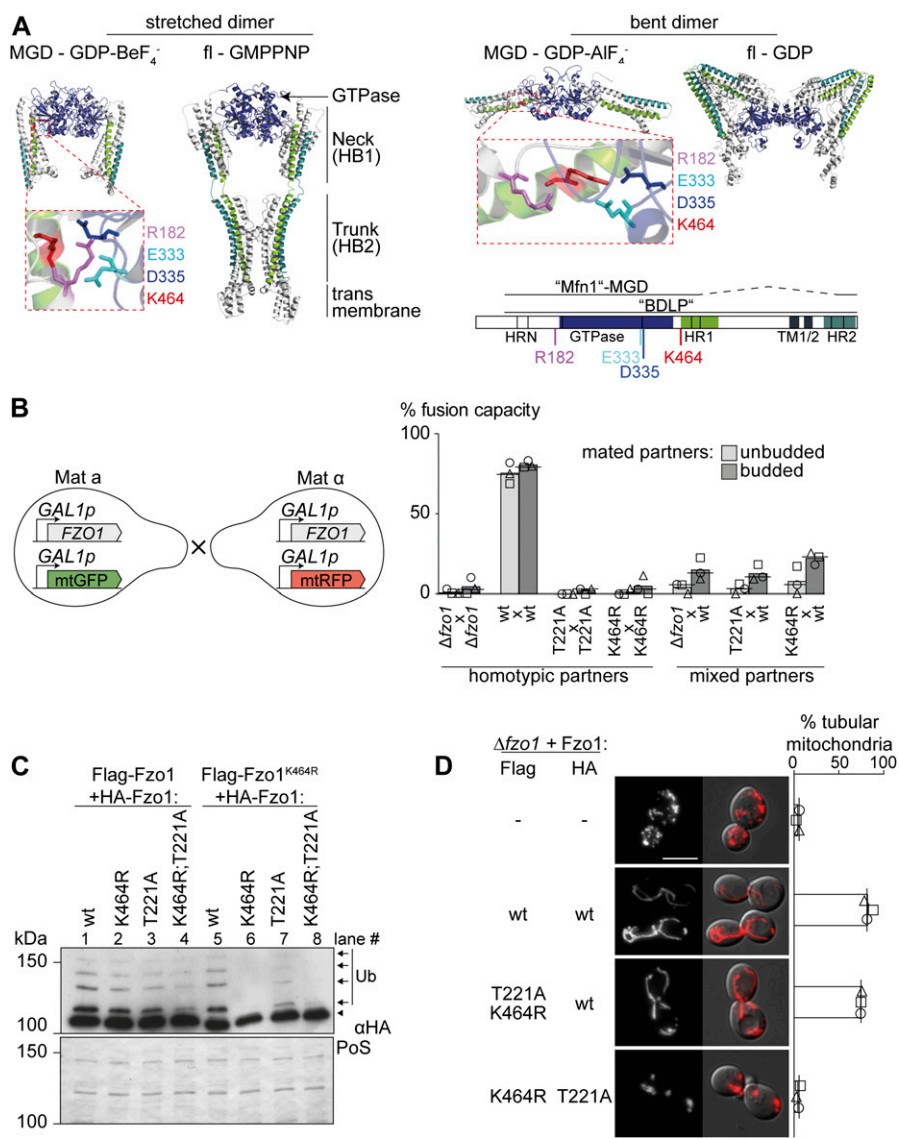

**Figure 1. Fzo1 ubiquitylation is not sufficient for mitochondrial fusion.**
**(A)** Crystal structure models of Fzo1. Left: stretched dimer. Fzo1 modelled on MFN1-MGD bound to GDP-BeF$_3^-$ and BDLP bound to GMPPNP. Right: bent dimer. Fzo1 modelled on GDP-AlF$_4^-$-bound MFN1-MGD and GDP-bound BDLP. Zoom-ins show residues proposed to form a salt bridge, displayed as sticks. Bottom right: Linear representation of the domain structure of Fzo1. **(B)** wt Fzo1 is required on each fusion partner to mediate fusion. Left: experimental setup of the mating assay for mitochondrial fusion. *FZO1* and *mtGFP* or *mtRFP* are expressed under the control of the repressible *GAL1* promoter in the two mating types a and α. Right: quantification of the fusion capacity after transcriptional repression by glucose, in budded or unbudded mated partners of Δ*fzo1* cells expressing the indicated Fzo1 variants. Three independent experiments were quantified (with more than 30 budded or unbudded events each), including mean (bars), median (lines), and individual experiments (circles, squares, and triangles). **(C)** Intermolecular cross talk rescues ubiquitylation in Fzo1$^{K464R}$ and Fzo1$^{T221A}$. Crude mitochondrial extracts from Δ*fzo1* cells expressing the indicated variants of Flag-Fzo1 and HA-Fzo1 were solubilized and analyzed by SDS–PAGE and immunoblotting using HA-specific antibodies. Unmodified and ubiquitylated forms of HA-Fzo1 are indicated by a black arrowhead or black arrows, respectively. Ubiquitylated forms of Fzo1 are labeled with Ub. **(D)** Fzo1 mutants permissive to its ubiquitylation fail to rescue mitochondrial fusion. Analysis of mitochondrial tubulation in Δ*fzo1* cells expressing the indicated Flag- or HA-tagged variants of Fzo1, co-expressing a mitochondrial-targeted mCherry plasmid. Cellular (Nomarski) and mitochondrial (mCherry) morphology were visualized by fluorescence microscopy. Three independent experiments were quantified (with more than 200 cells each), including mean (bars), median (lines), and individual experiments (circles, squares, and triangles). Scale bar: 5 $\mu$m. fl, full length; MGD, minimal GTPase domain; PoS, PonceauS staining; TM, transmembrane domain; HRN/HR1/HR2, heptad repeats.

structures between the C-terminal domains, proposing stabilization of a fusion-competent state of mitofusin (Koshiba et al, 2004; Franco et al, 2016).

Ubiquitin, an essential exchange currency for virtually all dynamic processes, was shown to be a key regulator of mitofusins (Escobar-Henriques, 2014; Escobar-Henriques & Joaquim, 2019). Ubiquitin is covalently attached to lysine residues of target proteins, via an enzymatic cascade operated by E1, E2, and E3 enzymes (Ciechanover, 2015; Yau & Rape, 2016). Deubiquitylases (DUBs), which remove ubiquitin chains, reverse ubiquitylation and offer possibilities for regulation (Clague et al, 2019). The ubiquitin-dedicated chaperone p97/Cdc48 is another important regulator of proteins modified by ubiquitin, also allowing remodeling of membrane proteins (Bodnar & Rapoport, 2017). Ubiquitylation of mitofusins is conserved from yeast to fly and mammals (Cohen et al, 2008; Ziviani et al, 2010; Anton et al, 2011; Rakovic et al, 2011). Fzo1 ubiquitylation is essential for OM fusion in yeast and is subject to a

tight regulation, for example, via a deubiquitylase cascade governed by Cdc48 (Anton et al, 2013; Chowdhury et al, 2018; Simoes et al, 2018; Goodrum et al, 2019). Moreover, ubiquitylation occurs downstream of self-oligomerization and GTP hydrolysis and requires the lysine 464 (Anton et al, 2011, 2013), a conserved and CMT2A disease-linked residue (Zuchner et al, 2004).

Here, to gain mechanistic insights into how mitofusins drive the process of OM fusion, we transferred structure- and *in organello*–based hypotheses into in vivo analyses of mitochondrial fusion capacity, using yeast cells. This was particularly relevant because the structural data on MFN1 lack the HB2 trunk, that is, lack information about the behavior of mitofusin proteins in their lipid context. We investigated the link between conformational changes and K464 dependence for Fzo1 ubiquitylation. We show that K464 is involved in a tripartite salt bridge essential for fusion and is only required after GTP hydrolysis. Moreover, ubiquitylated but fusion-incompetent intermediates of Fzo1 could be identified. This compelled a reassignment

for the role of Fzo1 ubiquitylation in the multistep process of mitochondrial fusion. Consistently, we could demonstrate that only ubiquitylated Fzo1 can be recognized and disassembled by Cdc48, which thereby promotes efficient and sustained fusion events.

# Results and Discussion

### Mitochondrial fusion requires lysine 464 in Fzo1 on both mitochondrial partners

The lysine residue 464 in Fzo1, which when mutated in MFN2 is causative of CMT2A, is essential for mitochondrial fusion, in yeast and in mammals (Fig S1A and B), and consequently for respiratory capacity (Fig S1C). Moreover, mutations of K464 revealed a stringent requirement for the presence of a lysine residue at this position (Fig S1C). K464 is also required for Fzo1 ubiquitylation (Fig S1D, compare lanes 1 and 2; [Anton et al, 2013]). However, we previously noted that co-expression of wild-type (wt) Fzo1 rescues ubiquitylation of Fzo1$^{K464R}$, suggesting complementation within the Fzo1 oligomer and clearly showing that the observed ubiquitylation is not conjugated on K464 (Fig S1D, compare lanes 2 and 6; [Anton et al, 2013]). Nevertheless, it was unclear if this oligomeric cross talk between Fzo1 molecules occurs in *cis* or in *trans*. Thus, to elucidate the exact role of K464 in the process of OM fusion, we first determined if it is required on both sides of the fusing partners. To this aim, we scored mitochondrial fusion capacity of cells expressing either wt or K464R variants of Fzo1, using a previously described mating assay (Nunnari et al, 1997). Co-localization of different mitochondrial markers indicates mitochondrial network mixing, and thus fusion capacity (Fig S1E and F). To avoid possible artifacts, we slightly modified the mating assay, by shutting off the expression of Fzo1 and of the mitochondrial fluorescent markers before mating, using the repressible promoter of *GAL1* (Fig 1B). As expected, homotypic reactions revealed the dependence on Fzo1 for mitochondrial fusion (compare [Δ*fzo1* × Δ*fzo1*] with [wt × wt]). Furthermore, the GTP hydrolysis dead variant Fzo1$^{T221A}$ also abolished fusion (Fig 1B; T221A × T221A; [Hermann et al, 1998]). Similarly, K464 was essential for mitochondrial fusion (Fig 1B, K464R × K464R), consistent with the tubulation and respiratory defects. Importantly, heterotypic (mixed) pairing confirmed the requirement of wt Fzo1 in both fusion partners (Fig 1B, Δ*fzo1* × wt), validating the modified mating assay. In addition, cells containing Fzo1$^{K464R}$ maintained a strong fusion defect even when paired up with cells expressing wt Fzo1 (Fig 1B, K464R × wt), similar to GTP hydrolysis mutants (T221A × wt). Of note, neither the K464R nor the T221A mutations had a dominant negative effect on mitochondrial tubulation when co-expressed with wt Fzo1 (Fig S1G). Together, these results show that K464 is needed on both fusion partners.

### Fzo1 ubiquitylation is necessary but insufficient for mitochondrial fusion

Similar to the mutations in K464, impairing GTP hydrolysis abolished Fzo1 ubiquitylation, which could be rescued by the presence of wt Fzo1 (Fig S1D, compare lanes 3 and 7). Even the double mutant

Fzo1$^{T221A;K464R}$ regained ubiquitylation in the presence of endogenous Fzo1 (Fig S1D, compare lanes 4 and 8). To further challenge this, we analyzed if co-expression of Fzo1$^{K464R}$ and Fzo1$^{T221A}$ would be sufficient to allow Fzo1 ubiquitylation. We, therefore, expressed differently tagged versions of Fzo1 (Flag or HA) in Δ*fzo1* cells, harboring the required combinations of T221A and K464R mutations. Strikingly, HA-Fzo1$^{T221A}$ was ubiquitylated when expressed in the presence of Flag-Fzo1$^{K464R}$ to similar levels as the double mutant in presence of the wt protein (Fig 1C, compare lanes 4 and 7). This shows that no wt Fzo1 is needed to achieve Fzo1 ubiquitylation. Nevertheless, co-expression of Flag-Fzo1$^{K464R}$ and HA-Fzo1$^{T221A}$ in Δ*fzo1* cells was not able to restore mitochondrial fusion (Fig 1D). Together, these results show that Fzo1 ubiquitylation is necessary but insufficient to permit mitochondrial fusion.

### Residues proximal to K464 are also required for Fzo1 ubiquitylation and functionality

Our results showed that even after rescue of ubiquitylation, Fzo1$^{K464R}$ mutants are still not capable of promoting mitochondrial fusion. Thus, despite confirming a critical function of K464, the reason thereof is certainly beyond Fzo1 ubiquitylation. K464 locates to a hinge region between the GTPase and the tightly packed neck region (HB1), critical for switches between the stretched and bent dimer conformations (Fig 1A; [Qi et al, 2016; Cao et al, 2017; Yan et al, 2018]), whose importance is underlined by the massive mapping of CMT2A mutations (Barbullushi et al, 2019). Moreover, the MFN1-MGD structures suggested the homologue of K464 to be mediating this structural dynamism, by being part of a salt bridge together with three additional amino acids in this region, partly also causative of CMT2A (Yan et al, 2018; Dankwa et al, 2019). In yeast, these correspond to the positively charged R182 and the negatively charged E333 and D335 (Fig 1A, zoom-ins, Fig S1B). Therefore, we analyzed their role for Fzo1 functionality in vivo. Among the negatively charged residues, we identified D335 as being stringently essential for Fzo1 activity (Fig S2A), where even its mutation to the likewise negatively charged glutamate did not rescue mitochondrial tubulation (Fig S2B). Similarly, mutation of R182 to lysine, that is, another positive residue, completely impaired mitochondrial tubulation and Fzo1 ubiquitylation (Fig S2C). Strikingly, even swapping lysine and arginine at residues R182 and K464 abolished Fzo1 functionality (Fig S2C). Therefore, despite their similar position and orientation, R182 and K464 could not be functionally exchanged. In sum, we identified the residues R182 and D335 in Fzo1 as being required, like K464, for Fzo1 ubiquitylation and mitochondrial fusion.

### Dynamic interplay at the hinge region between HB1 and GTPase domain is essential for Fzo1 activity

The observation that K464, R182, and D335 are essential, that is, three residues proposed to form salt bridges, raised the question whether the different possible configurations of the salt bridge are required during different stages. Indeed, according to Fzo1 modelled to the MFN1-MGD crystal structures, the two positively charged R182 and K464 undergo noticeable changes in orientation and distance to D335, depending on the nucleotide state (Fig 2A). R182 is close to D335 in Fzo1-MGD bound to GDP-BeF$_3^-$ and further away in

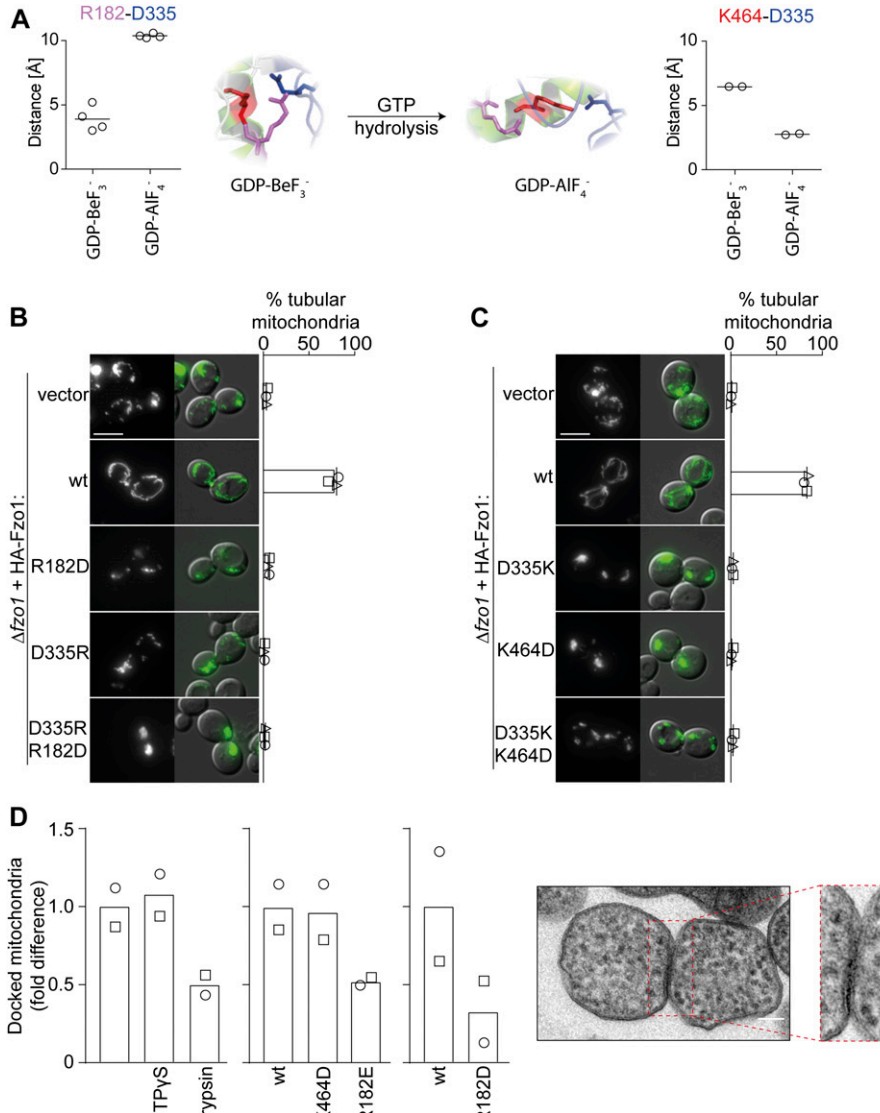

**Figure 2. Double salt bridge swaps block mitochondrial fusion.**
**(A)** Alternation of D335 positioning. Fzo1-MGD modelled on MFN1 bound to GDP-BeF$_3^-$ (left) and GDP-AlF$_4^-$ (right) and corresponding distance predictions between all charged ends of D335 and either R182 or K464, resulting in either four or two measurements, respectively. **(B, C)** Single charge swaps do not rescue mitochondrial fusion. Mitochondrial morphology of Δ*fzo1* cells expressing the indicated HA-Fzo1 variants, co-expressing a mitochondrial-targeted GFP plasmid, analyzed as in Fig 1D. Scale bar: 5 μm. **(D)** In vitro analysis of mitochondrial docking sites. Mitochondria were purified from *ugo1-2* cells (left) or from Δ*fzo1 ugo1-2* cells expressing HA-Fzo1, HA-Fzo1$^{K464D}$, HA-Fzo1$^{R182E}$ (middle), or HA-Fzo1$^{R182D}$ (right) and analyzed by TEM for docked events. Mitochondrial tethering was performed in the presence of 1 mM GTPγS or mitochondria were treated with 0.5 μg/ml trypsin before tethering, as indicated (left). At least 900 (left), 1,000 (middle), or 650 (right) mitochondria from two independent experiments were quantified, as described in Fig S2F, including mean (bars) and individual experiments (circles and squares). Example of a mitochondrial docking event (far right). Scale bar: 100 nm.

Fzo1-MGD bound to GDP-AlF$_4^-$. Vice versa, K464 is closer to D335 in the GDP-AlF$_4^-$ than in the GDP-BeF$_3^-$ nucleotide state (Fig 2A). This suggested that R182 and K464 could be alternating in salt bridge interactions with D335 (Yan et al, 2018). To analyze the importance of the two putative alternating salt bridges between D335 and either R182 or K464, we tested if pair-wise charge swapping between each of them would be sufficient to rescue Fzo1 functionality. First, we tested a charge exchange between R182 and D335. However, the Fzo1$^{R182D;D335R}$ swap variant could not restore mitochondrial tubulation or Fzo1 ubiquitylation, when compared with wt Fzo1 (Figs 2B and S2D). Similarly, a charge swap between D335 and K464 did not rescue mitochondrial tubulation or Fzo1 ubiquitylation (Figs 2C and S2E, left panel). Nevertheless, as previously reported (De Vecchis et al, 2017), in the strain background W303 the Fzo1$^{K464D;D335K}$ variant could partially rescue Fzo1 ubiquitylation (Fig S2E, right panel). These results show that a salt bridge between the negative residue D335 and either one of the positive residues R182 or K464 alone is not sufficient to mediate Fzo1 activity.

Next, we questioned whether R182–D335 and K464–D335 interactions would reflect previously identified "docked" and "tethered" OM fusion states, respectively (Hoppins et al, 2009; Brandt et al, 2016). The change from a "tethered" to a "docked" state of the fusion complex was defined by an increase in the contact area between apposing mitochondria, leading to increased membrane deformations (Hoppins et al, 2009). Therefore, we analyzed the tethered and docked *status* between isolated mitochondria, harboring either Fzo1$^{R182E}$ or Fzo1$^{K464D}$ via transmission electron microscopy (Fig S2F). However, the number of docked mitochondria was very low (Fig S2F). To clearly observe differences between the R182E- and K464D-mutant variants, mitochondria arrested at the docking stage were used, as presented in *ugo1-2*–mutant cells (Fig S2F; [Hoppins et al, 2009]). This prevents downstream disassembly

of docked fusion complexes, thus allowing to test if the mutants Fzo1$^{R182E}$ and Fzo1$^{K464D}$ reach this stage or are instead arrested before docking. First, we confirmed that mitochondrial docking is independent on GTP hydrolysis, acting as a positive control, being, however, sensitive to trypsinized mitochondria, acting as a negative control (Fig 2D; [Hoppins et al, 2009]). Subsequently, we could not observe differences between the wt and the K464D variant in the relative number of docked mitochondria, consistent with its requirement only after GTP hydrolysis (Fig 2D). Strikingly, and in contrast, cells expressing the mutant variant Fzo1$^{R182E}$ or Fzo1$^{R182D}$ were severely impaired in reaching the docking state (Fig 2D). Together, our results emphasize the importance of both salt bridges at different stages of the fusion process (Fig 2D).

### A trilateral salt bridge between K464, D335, and R182 mediates OM fusion

The functional impairment upon mutations in K464, R182, and D335 or upon pair-wise exchange between them suggests that the interplay between all three residues could be stringently required for Fzo1 functionality. Thus, we predicted that only a triple charge swap would restore the capacity for dynamically alternating salt bridge interactions between the residue in position 335 with the ones in positions 464 or 182. Consistently and remarkably, the variant Fzo1$^{R182E;D335K;K464D}$, possessing a simultaneous charge swap of all three residues, allowed mitochondrial tubulation (Fig 3A) and Fzo1 ubiquitylation (Fig 3B). Next, we sought out to confirm the capacity of the triple swap mutant in mediating membrane fusion. Strikingly, budded zygotes of mated cells harboring the triple salt bridge mutations reached almost wt-like levels (Fig 3C). Albeit with

decreased efficiency, this confirms the functionality of the triple swap variant of Fzo1. These results further emphasize the requirements for several rounds of conformational switches during the OM fusion process (Brandt et al, 2016; Rocha et al, 2018), consistent with the behavior of atlastins (Liu et al, 2015). In contrast, simultaneous mutation of R182, D335, and K464 to the neutrally charged residue alanine, which prevents all possible interactions, abolished mitochondrial tubulation (Fig 3D). Together, our results demonstrate that the presence of electrostatic interactions at the hinge region between the GTPase and HB1 is essential for Fzo1 function. Moreover, we show that trilateral and dynamic salt bridge interactions are required during the fusion process.

### Fzo1 ubiquitylation on fusion-incompetent variants of Fzo1 is not regulated by Cdc48

Next, we sought out to further understand the role of Fzo1 ubiquitylation in OM fusion, profiting from our identification, on the one side, of fusion-competent and, on the other side, ubiquitylated but fusion-incompetent mutant forms of Fzo1 (e.g., T221A and K464R in the presence of wt Fzo1). In fact, ubiquitylated but fusion-incompetent Fzo1 is likely not able to undergo conformational changes that are rescued in the Fzo1 triple salt bridge mutant. We hypothesized that the ubiquitin-specific chaperone Cdc48 would not recognize the fusion-incompetent Fzo1 forms because of lack of these conformational changes. First, we compared the response to Cdc48 of HA-Fzo1, HA-Fzo1$^{T221A;K464R}$, and the corresponding single mutants (Fig 4A). This experiment was performed in wt cells, that is, in the presence of endogenous Fzo1, to complement ubiquitylation in the mutant variants. As expected, for wt, Fzo1 ubiquitylation was significantly reduced

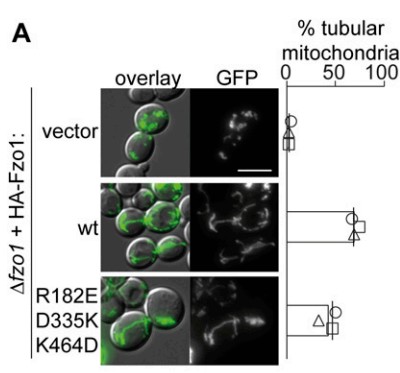

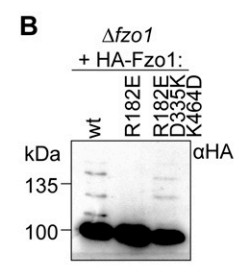

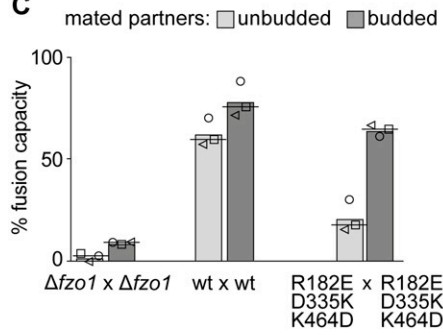

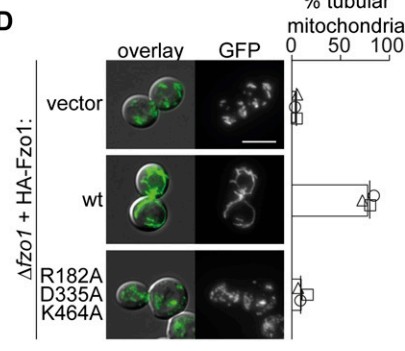

**Figure 3. Triple salt bridge swap rescues mitochondrial fusion.**
**(A, D)** Fusion is rescued by a double positive charge swap in (A) but not by the presence of neutral amino acids in (D). Mitochondrial morphology and quantification of Δ*fzo1* cells expressing the indicated Fzo1 variants, co-expressing a mitochondrial-targeted GFP plasmid, analyzed as in Fig 1D. Scale bar: 5 μm. **(B, C)** Triple salt bridge swap between residues in positions 182, 335, and 464 rescues Fzo1 ubiquitylation in (B) and fusion capacity in (C). The indicated Fzo1 mutant variants were analyzed for ubiquitylation as in Fig 1B and for fusion capacity as in Fig 1D. PoS, PonceauS staining.

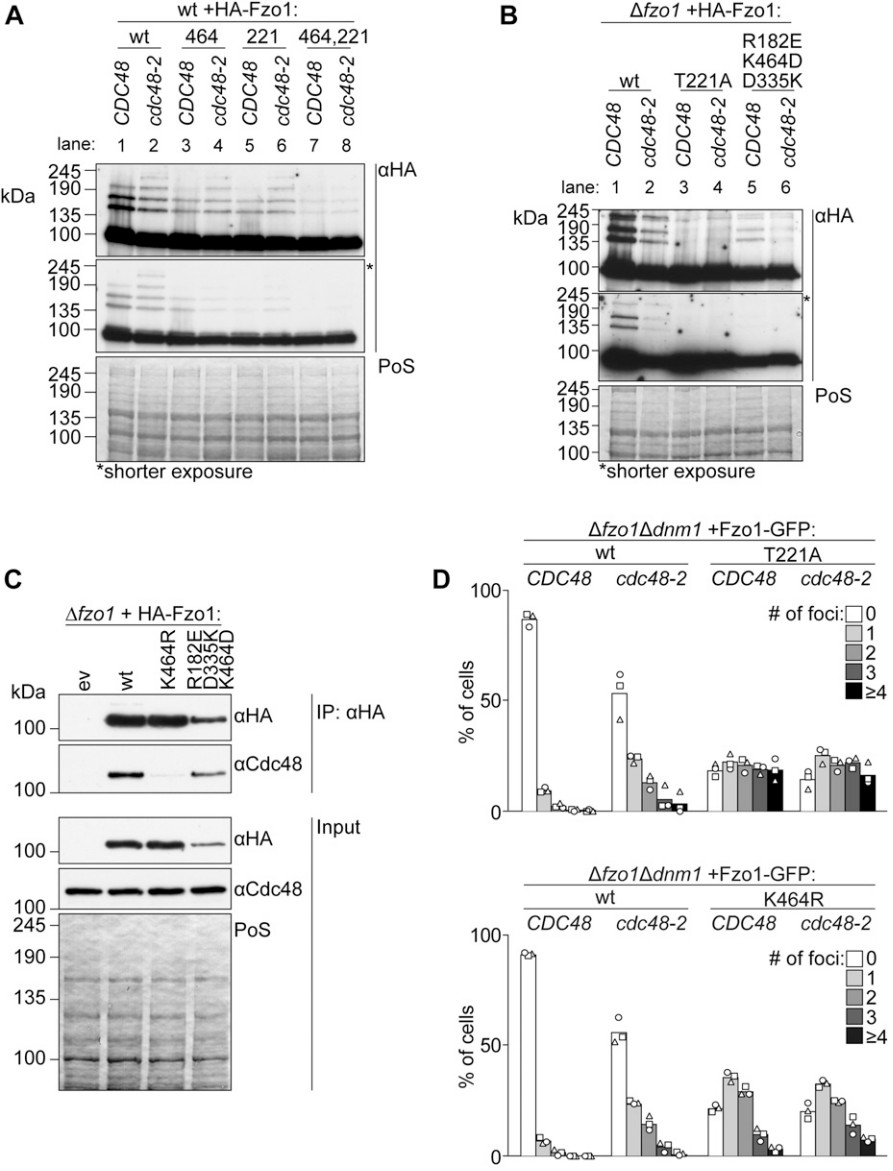

**Figure 4. Fusion-incompetent ubiquitylated Fzo1 is insensitive to Cdc48.**
**(A, B)** Ubiquitylation of the indicated HA-tagged Fzo1 mutant variants, expressed in wt and *cdc48-2* cells in (A) or in *Δfzo1* and *Δfzo1cdc48-2* cells in (B). Total cell extracts were prepared and analyzed by SDS–PAGE and immunoblotting, using HA-specific antibodies. **(C)** Analysis of Cdc48-Fzo1 co-immunoprecipitation. The indicated HA-Fzo1 variants were expressed in *Δfzo1* cells. Crude mitochondrial extracts were solubilized, subjected to co-immunoprecipitation, and analyzed by SDS–PAGE and Western blot using HA- and Cdc48-specific antibodies. **(D)** Localization of indicated Fzo1-GFP variants, expressed in *Δfzo1Δdnm1* and *Δfzo1Δdnm1cdc48-2* cells. Fzo1-GFP was co-expressed with Su9-mCherry. Fzo1-GFP *foci* were quantified as shown in Fig S3 in at least 100 cells showing a tubular mitochondrial network, including mean (bars) and individual experiments (circles, squares, and triangles). PoS, PonceauS staining.

in *cdc48-2* cells (Fig 4A, compare lanes 1 and 2). In contrast, the T221A, K464R, and double mutant variants were insensitive to Cdc48 impairment (Fig 4A, compare lanes 3 and 4, 5 and 6, and 7 and 8). This indicates that regulation by Cdc48 only occurs on fusogenic active forms of Fzo1. Thus, we wondered whether the partially functional triple swap mutant is recognized by Cdc48. Indeed, HA-Fzo1$^{R182E;D335K;K464D}$ was sensitive to Cdc48, whereas the non-functional HA-Fzo1$^{T221A}$ was not (Fig 4B). Consistently, only wt and Fzo1$^{R182E;D335K;K464D}$, but not nonfunctional Fzo1$^{K464R}$, interact with Cdc48 (Fig 4C). Given that Cdc48 acts as a segregase (Cooney et al, 2019; Twomey et al, 2019), we hypothesized that impairment of Cdc48 function would lead to the accumulation of Fzo1 at stalled mitochondrial fusion sites. To specifically examine the localization of Fzo1, we had to overcome the aggregation of mitochondria present in *cdc48-2*–mutant cells. Thus, Fzo1-GFP was analyzed in mitochondria tubulated by deletion of *DNM1*. Indeed, we could find

an increase in Fzo1-GFP *foci* in *cdc48-2*–mutant cells, when compared with wt cells (Figs 4D and S3). Furthermore, as expected, expression of Fzo1$^{T221A}$-GFP or Fzo1$^{K464R}$-GFP led to the formation of *foci* even in the presence of wt Cdc48 (Figs 4D and S3). This is consistent with the capacity of both mutant variants to tether and dock mitochondria (Fig 2D; [Anton et al, 2011]) and form clusters (Brandt et al, 2016). Together, these results support a role of Cdc48 in segregating Fzo1 aggregates, after GTP hydrolysis, dependent on Fzo1 ubiquitylation.

In sum, we uncover an original regulatory mechanism of ubiquitin-dependent membrane fusion. Indeed, first, our results indicate that Cdc48 only acts on fusion-competent variants of Fzo1, after membrane merging, by clearing ubiquitylated Fzo1 from fusion sides. Second, we show that ubiquitin recognition by Cdc48 depends on dynamically alternating tripartite salt bridge formations, likely stabilizing conformational changes driven by GTP binding and hydrolysis.

## Mechanism of outer mitochondrial membrane (OMM) fusion

Our results allow the proposal of an updated model for the multiple step process required for mitochondrial fusion, integrating into previous knowledge the role of Cdc48 and of an alternating salt bridge (Figs 5, and S4A and B). It is composed of one negative residue (D335) dynamically interacting with two positive ones (K464 and R182). We propose a critical role of the trilateral salt bridge in stabilizing the two conformational stages in mitofusins, before and after GTP hydrolysis, thus actively assisting the fusion process. Ultimately, understanding how mitofusins regulate mitochondrial morphology could contribute to therapeutic interventions of CTM2A, which is still incurable.

First, the *cis*-dimers present at the mitochondrial surface (1) further oligomerize *in trans*, allowing mitochondrial tethering (2), independently of GTP hydrolysis (Anton et al, 2011; Cohen et al, 2011), but dependent on a salt bridge interaction between D335 and R182. Second, bending of the Fzo1 oligomers, driven by GTP hydrolysis (3), shifts the salt bridge from R182 to K464. Moreover, recurring cycles of GTP loading and hydrolysis (4) are required to allow OM fusion (Brandt et al, 2016). However, ubiquitylation only occurs after GTP hydrolysis (Fig 1B, see lane 3; [Anton et al, 2011; Cohen et al, 2011]). Therefore, after one/several rounds of GTP hydrolysis, Fzo1 is ubiquitylated (5). However, ubiquitylation is necessary but not sufficient for OM fusion. Indeed, after ubiquitylation of Fzo1, merging of the two apposing membranes occurs

(6), which can then evolve to total fusion of the OM (7) (Brandt et al, 2016). Finally, Fzo1 ubiquitylation can then be regulated by Cdc48, thus allowing controlled and sustained fusion events (8). We propose that Cdc48 disassembles the tethering complex, in analogy to the role of NSF in SNARE-mediated fusion (Ryu et al, 2016; Huang et al, 2019), allowing Fzo1 recycling for new rounds of GTP binding.

# Materials and Methods

## Yeast strains and growth media

Yeast strains, except Δ*fzo1* (W303) and *ugo1-2* (W303) (Hoppins et al, 2009), are isogenic to the S288c (Euroscarf). They were grown according to standard procedures to the exponential growth phase at 30°C (unless stated otherwise) on yeast-extract peptone (YP) or synthetic complete (SC) media supplemented with 2% (wt/vol) glucose (D), 3% (wt/vol) glycerol, or 2% (wt/vol) galactose.

## Cell lines and cultivation

Immortalized $MFN2^{-/-}$ homozygous knockout MEFs (Chen et al, 2003) were cultured at 37°C and 5% $CO_2$ in a humidified incubator in DMEM–GlutaMAX containing 4.5 g/l glucose (#61965026; Thermo Fisher Scientific) supplemented with 1 mM sodium pyruvate (#11360039;

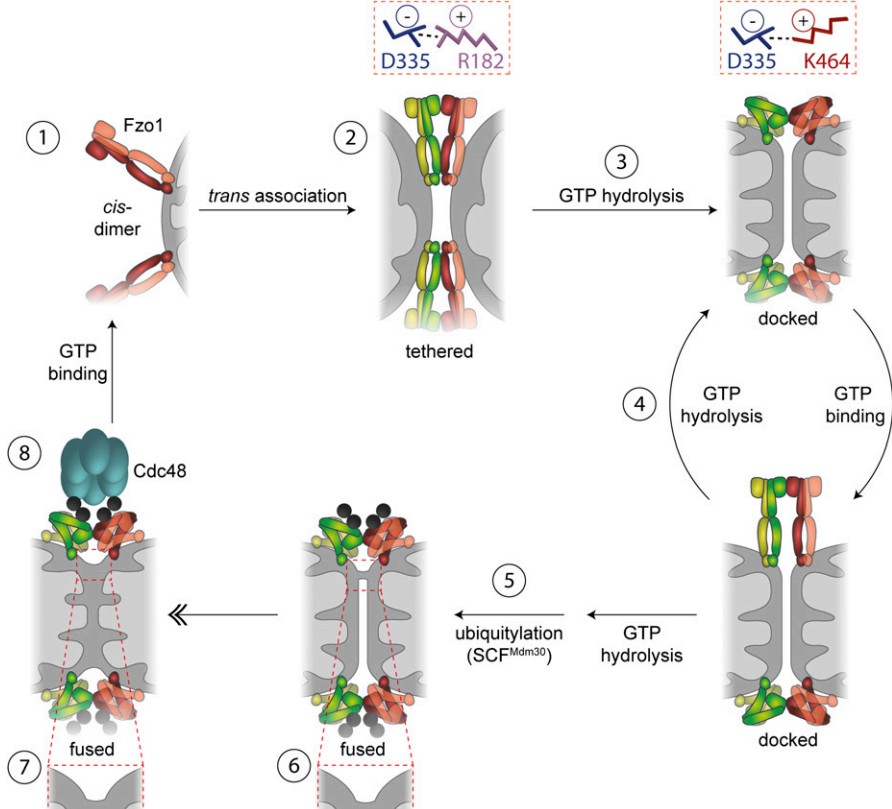

**Figure 5. Integrated model for mitochondrial OM fusion.**

Model for OM fusion. GTP-bound Fzo1 dimers localize at the OMM (1). Fzo1 *trans* association leads to formation of the tethering complex, which depends on dynamic salt bridge interactions (2). GTP hydrolysis shifts the salt bridge from R182 to K464 and thereby drives conformational changes on Fzo1 (3) eventually promoting membrane curvature and formation of the docked stage. Recurring cycles of GTP binding and hydrolysis (4) allow membrane approximation and ubiquitylation of Fzo1 by SCF[Mdm30] (5), eventually allowing local lipid merging (6), which rapidly expands for complete fusion of the two OMs (7). After membrane merging, Fzo1 ubiquitylation is controlled by Cdc48, possibly leading to complex disassembly (8).

Thermo Fisher Scientific), 100 $\mu$M nonessential amino acids (#11140035; Thermo Fisher Scientific), and 10% FBS (S0115; Biochrom). The cells were transiently transfected using Lipofectamin 2000 (#11668; Thermo Fisher Scientific). Lipofectamin 2000 was incubated 5 min at RT in Opti-MEM (#31985070; Thermo Fisher Scientific) before adding 1 $\mu$g plasmid per six-well plate and incubation for 15 min at RT. Transfection mix was added drop wise to plated cells. Transient transfection was performed for 48 h, whereby the medium was exchanged after 24 h.

### Plasmids

The following plasmids were previously described: mouse MFN2-Flag (Hoppins et al, 2011), pRS315 (plasmid # (p) 7) and pRS316 (p8) (Sikorski & Hieter, 1989), pRS415 (p132) (Simons et al, 1987), HA-Fzo1 (p10) and HA-Fzo1$^{T221A}$ (p34) (Anton et al, 2011), HA-Fzo1$^{K464R}$ (p14) (Anton et al, 2013), Flag-Fzo1 (p11) (Escobar-Henriques et al, 2006), and 3xMyc-Fzo1 under the control of the *GAL1* promoter (p350) (Hermann et al, 1998). Equally, mitochondrial matrix targeted (mt) GFP encoded on pYX142 (p70) and pVT100 (p68), and on pYX113 under the control of the *GAL1* promoter (p488); mtdsRed on pVT100 (p69); and mtRFP on pYX113 under the control of the *GAL1* promoter (p487) were all previously described (Westermann & Neupert, 2000). GFP-tagged Fzo1 (p86) was cloned by first replacing the Fzo1 coding sequence from p10 with a Fzo1 coding sequence without stop codon using XhoI and SalI. Second, the GFP coding sequence, including a flexible linker between *FZO1* and *GFP* (CGG ATC CCC GGG TTA ATT AAC) was cloned into this vector using SalI and XbaI. Mitochondrial-targeted mCherry (p421) was cloned into pRS413, under the control of the promoter of *Translational elongation factor EF-1α* (*TEF1*) and the terminator of *Cytochrome c* (*CYC1*), with BamHI and XhoI. The N-terminal mitochondrial targeting site of Su9 was subsequently cloned into the same vector using XbaI and BamHI. Plasmids encoding point mutants in HA-Fzo1 (p327: K464Q, p402: K464N, p403: K464A, p404: K464E, p406: K464F, p411: K464D, p447: K464W, p412: E333R, p415: D335K, p539: D335E, p541: D335V, p540: D335A, p552: R182E, p555: R182K, p600: R182D, p125: T221A; K464R, p414: E333R; D335R, p416: D335K; K464D, p601: R182D; D335K, p556: R182K; K464R, p553: R182E; D335K; K464D and p642: R182A; D335A; K464A), Flag-Fzo1 (p473: K464R and p448: T221A; K464R), HA-Fzo1-GFP (p273: K464R and p808: T221A), or 3xMyc-Fzo1 (p542: T221A and p543: K464R) were generated by point mutagenesis, in the corresponding plasmids above described (p10, p11, p86, and p350, respectively). The plasmid encoding HA-Fzo1$^{R182E;D335K;K464D}$ under the control of the *GAL1* promoter (p641) was amplified from the plasmid encoding HA-Fzo1$^{R182E;D335K;K464D}$ under the control of the *FZO1* promoter (p553) and cloned with SalI and XhoI into the same sites of plasmid encoding 3xMyc-Fzo1 (p350).

### Antibodies

The antibodies anti-HA (1:1,000 in 5% milk in TBS; #11867423001; Roche), anti-Flag M2 (1:1,000 in 5% milk in TBS; F3165; Merck), and anti-Cdc48 (gifted by T Sommer) were used in this study.

### Spot tests

For growth assays, Δ*fzo1* cells expressing different Fzo1 plasmids were generated by tetrad dissection. Serial 1:5 dilutions of exponentially growing cells using a starting OD$_{600}$ of 0.5 were spotted on

YP or SC media containing glucose or glycerol and were grown at 30°C.

### Total cell extraction for Fzo1 steady state levels and ubiquitylation

For analysis of protein steady state levels and ubiquitylation, total proteins from three OD$_{600}$ exponentially growing cells were resuspended in 1 ml of ice-cold water with 260 mM NaOH and 7.5% $\beta$-mercaptoethanol and incubated on ice for 15 min. Trichloroacetic acid (TCA) was added to a final concentration of 6.5%, and the suspension was incubated for 10 min on ice. The suspensions were centrifuged at 16,100$g$ for 10 min at 4°C. The supernatant was aspirated and the pellet was dried. The pellet was resuspended in Hydroxy urea buffer (8M Urea, 5% SDS, 200 mM Tris, pH 6.8, 0.01% bromophenol blue, and freshly added 100 mM DTT). Samples were heated to 65°C for 10 min (shaking) before analysis by SDS–PAGE and immunoblotting.

### Crude membrane extraction for Fzo1 ubiquitylation

Crude membrane extracts were essentially performed as described before (Schuster et al, 2018). 30 OD$_{600}$ of yeast cells grown in SCD media to the exponential growth phase were disrupted with glass beads (0.4–0.6 $\mu$m) in TBS with 6.6 mM PMSF and cOmplete Protease Inhibitor Cocktail (Roche). After centrifugation, at 16,000$g$ for 10 min, the pellet (containing crude membranes) was resuspended in 20 $\mu$l solubilisation buffer (0.2% NG310 [Anatrace] in TBS) for rotating at 4°C for 1 h. The reaction was stopped by adding 2× Laemmli buffer. After incubation at 45°C for 20 min (shaking), the samples were analyzed by SDS–PAGE and immunoblotting.

### Immunoprecipitation for analysis of Fzo1 ubiquitylation

Crude membranes were extracted and solubilized from 100 OD$_{600}$ exponentially growing yeast cells as described above but in 500 $\mu$l solubilisation buffer (Schuster et al, 2018). Solubilized extracts were centrifuged for 5 min at 16,100$g$ and 4°C. 4% of the supernatant was kept as input control, the remaining 96% of the supernatant was incubated with 25 $\mu$l HA-coupled beads (EZview Red Anti-HA Affinity Gel, E6779; Sigma-Aldrich) overnight rotating at 4°C. Three washes were performed with 0.2% NG310 in TBS. HA-Fzo1 was eluted in 50 $\mu$l Laemmli buffer for 20-min shaking at 45°C and analyzed by SDS–PAGE and immunoblotting.

### Analysis of the interaction between HA-Fzo1 and Cdc48

Physical interactions between Cdc48 and Fzo1 were analyzed as previously described (Simoes et al, 2018). Briefly, 160 OD$_{600}$ of yeast cells grown in complete media to the exponential growth phase were disrupted with glass beads (0.4–0.6 $\mu$m) in TBS. After centrifugation at 16,000$g$ for 10 min, the crude membrane fraction was solubilized using 0.2% NG310 for 1 h rotating at 4°C. HA-Fzo1 was immunoprecipitated using Flag-coupled beads (Sigma-Aldrich) rotating overnight at 4°C. Beads were washed three times with 0.2% NG310 in TBS and the precipitated protein was eluted in Laemmli buffer for 20-min shaking at 40°C. 4% of the input and 50%

of the eluate fractions were analyzed by SDS–PAGE and immuno-blotting, using HA-specific and Cdc48-specific antibodies.

## Mitochondrial morphology

Yeast strains were transformed with mitochondrial-targeted GFP or mCherry, grown on YPD or SC media to the exponential phase, and analyzed as described (Escobar-Henriques et al, 2006) by epifluorescence microscopy (Axioplan 2; Carl Zeiss MicroImaging, Inc.) using a 63× oil-immersion objective. Images were acquired with a camera (AxioCam MRm; Carl Zeiss MicroImaging, Inc.) and processed with Axiovision 4.7 (Carl Zeiss MicroImaging, Inc.). Quantifications of mitochondrial morphology are depicted as mean (bars), median (line), and individual replicates (circles, squares, and triangles), from three independent experiments with at least 200 cells.

MEF cells transiently transfected with the indicated MFN2 variants were plated on cover slips and incubated with 500 mM MitoTracker CMXRos (M7512; Thermo Fisher Scientific) for 1 h, at 37°C. The cells were washed twice with PBS and fixed with 3.7% paraformaldehyde for 20 min at 37°C. The fixed cells were dehydrated with 0.1% Triton diluted in PBS for 15 min at RT and blocked with 2% BSA for 1 h at RT. Primary antibody decoration (anti-FLAG M2, 1:1,000) was performed for 1 h at RT. Cover slides were washed twice with PBS for 15 min and subsequently decorated with the secondary antibody (Alexa Fluor 488 antimouse (H+L), A-11001; Invitrogen) and 1 μg/ml DAPI (#62248; Thermo Fisher Scientific) for 1 h at RT. Cover slides were washed twice for 15 min and mounted using ProLong Gold (P36934; Thermo Fisher Scientific). At least 75 cells were imaged and processed as described above.

## Modelling of Fzo1 and MFN2

Structural models of Fzo1 were created using i-Tasser (Roy et al, 2011). Fzo1 in a membrane context (amino acids 61–856) was modelled on BDLP bound to 5′-Guanylyl imidodiphosphate (GMPPNP) (Protein Data Bank Identifier [PDB ID] 2W6D; c score −2.41) and GDP (PDB ID 2J69; c score −0.21) (Low & Löwe, 2006; Low et al, 2009). Fzo1-MGD (amino acids 61–491; flexible linker [GSGSGSGGS]; 826–856) was modelled on mammalian MFN1 bound to GTP (PDB ID 5GNS; c score −0,61), GDP-BeF$_4^-$ (PDB ID 5YEW; c score −1.17), GDP-AlF$_3^-$ (PDB ID 5GOM; c score −1.24), and GDP (PDB ID 5GNT; c score −0.80) (Qi et al, 2016; Cao et al, 2017; Yan et al, 2018). MFN2-MGD (amino acids 1–385; flexible linker [GSGSGSGGS]; 713–757) was modelled on MFN1 bound to GDP-BeF$_4^-$ (PDB ID 5YEW; c score −0.64) (Yan et al, 2018). The indicated c scores range from −5 to +2, where a more positive score reflects a model of better quality. The obtained structure models were processed using PyMOL (Version 2.0 Schrödinger, LLC). Distance estimations were calculated using PyMOL.

## Isolation of mitochondria for electron microscopy

Mitochondria were extracted based on (Meeusen et al, 2004). 1000 OD$_{600}$ of yeast cells, grown to exponential phase in YPD supplemented with 3.5% ethanol, were harvested by centrifugation. Cell walls were removed by incubation in 50 mM β-mercaptoethanol in 0.1M Tris, pH 9.4, for 20 min at 30°C, 90 rpm, and subsequent incubation in 3 mg/ml lytic enzyme (ICN) in 1.2M sorbitol for 30 min at

30°C, 90 rpm. Spheroplasts were centrifuged at 1,500g for 5 min at 4°C and washed once with 1.2M Sorbitol to remove lytic enzyme. Spheroplasts were resuspended in ice-cold mitochondria isolation buffer (NMIB) (0.6M sorbitol, 5 mM MgCl$_2$, 50 mM KCl, 100 mM KOAc, and 20 mM Hepes, pH 7.4) and homogenized using a tight dounce on ice 50 times. Unlysed cells and debris were removed from extracts by centrifuging at 3,000g for 5 min at 4°C. Enriched mitochondria were pelleted by centrifuging the supernatant at 10,000g for 10 min at 4°C. Mitochondria-enriched pellets were resuspended in NMIB to a final concentration of 10 mg/ml. Mitochondrial tethering was induced by incubation in stage 1 buffer (20 mM PIPES KOH, pH 6.8, 150 mM KOAc, 5 mM MG(OAc)$_2$, and 0.6M sorbitol) for 30 min at 4°C. When indicated, 1 mM GTPγS in stage 1 buffer was added for 30 min to the tethering reaction or, instead, mitochondria were treated with 50 μg/ml trypsin before the tethering reaction.

## Electron microscopy

Extracted mitochondria were treated based on (Unger et al, 2017), but fixed in suspension using 1.5% glutaraldehyde, 3% formaldehyde, and 2.5% sucrose in 0.1M sodium cacodylate buffer o/n at 4°C. Mitochondria were spun down into a pellet at 13,000g in a 1.5-ml microfuge tube. The fragile pellet was washed carefully three times with ddH$_2$O and postfixed with 1% osmium tetroxide for 1 h at 4°C. The pellet was washed four times with ddH$_2$O and incubated in 0.5% uranyl acetate overnight at 4°C. The pellet was washed three times in ddH$_2$O and embedded in 2% low-melting agarose, which was cut into small pieces of 1-mm edge length using a razor blade. Agar pieces were dehydrated for 15 min using ascending ethanol concentrations of 50%, 70%, 90%, 2× 100%, and 2× propylene oxide at 4°C. Pieces were infiltrated with Epon/propylene oxide 1:1 overnight at 4°C and pure Epon for 6 h at RT and embedded into BEEM capsules with conical tip (#69913-01; Science Services) and cured for 48 h at 60°C. Images were acquired using a OneView 4K camera (Gatan) mounted on a Jem-2100Plus (Jeol) transmission electron microscope operating at 200 kV. Large montages of 100 images were acquired using SerialEM (Mastronarde, 2003).

## Analysis of tethering and docking events

Mitochondria were quantified as tethered when the contact between mitochondria could be identified by distinct membrane contact and changes of membrane curvature. Mitochondria were quantified as docked when this contact site further extended to over at least one-third of the diameter of the mitochondria. In addition, contact sites were only counted as docked if changes in the membrane curvature were visible, that is, a flat contact between two parallel membranes of the opposing mitochondria. Mitochondria with a diameter smaller than 100 nm or larger than 1 μm were excluded from quantification.

## Mating assay for assessment of fusion capacity

Analysis of mitochondrial fusion capacity was essentially performed as described (Hermann et al, 1998; Fritz et al, 2003). Exponentially growing cells of opposite mating types (BY4741 and BY4742),

expressing indicated Fzo1 variants and mitochondrial matrix targeted (mt) GFP or RFP, respectively, from either endogenous promoters, the *ADH1* promoter or the repressible *GAL1* promoters, as indicated, were mixed for 4 h at 30°C in YPD. When proteins were expressed under the control of their endogenous or a ubiquitous promoter, cells were grown in SCD and mated in YPD. When proteins were expressed under the control of the promoter of *GAL1*, cells were cultured in SC with 2% Raffinose, supplemented with 2% galactose for 1 h to induce Fzo1 expression and subsequently supplemented 2% glucose for 1 h before mating, to stop Fzo1 expression. Fluorophore co-localization was analyzed by fluorescence microscopy.

## Supplementary Information

## Acknowledgements

We would like to thank J Nunnari for the *ugo1-2*–mutant strain and for the plasmids pvt100-mtGFP and pvt100-mtRFP, B Westermann for the plasmid pYX142-mtGFP and for the plasmids pYX113-mtGFP and pYX113-mtRFP under the *GAL1* promoter, J Shaw for the pRS415-3xMyc-Fzo1 plasmid, and T Sommer for the Cdc48 antibody. We are grateful to T Tatsuta and E Rugarli for critical reading of the manuscript and to G Praefke for stimulating discussions. This work was supported by grants of the Deutsche Forschungsgemeinschaft (ES338/3-1, Collaborative Research Centre 1218 TP A03; to M Escobar-Henriques), the Center for Molecular Medicine Cologne (CAP14, to M Escobar-Henriques), was funded under the Institutional Strategy of the University of Cologne within the German Excellence Initiative (Zukunft-skonzept [ZUK] 81/1, to M Escobar-Henriques), and benefited from funds of the Faculty of Mathematics and Natural Sciences, attributed to M Escobar-Henriques.

### Author Contributions

V Anton and M Escobar-Henriques designed the study and wrote the manuscript, with input from all authors.
V Anton performed most experiments and prepared the figures.
I Buntenbroich performed many experiments and prepared the figures.
R Schuster performed some experiments and prepared the figures.
T Simões performed the Fzo1- Cdc48 co-immunoprecipitations.
S Altin performed the MFN2 experiment.
F Babatz performed transmission electron microscopy experiments, under the supervision of A Schauss.
G Calabrese and J Riemer provided the Su9-mCherry plasmid.
M Escobar-Henriques coordinated the study.

### Conflict of Interest Statement

The authors declare that the research was conducted in the absence of any commercial or financial relationships that could be construed as a potential conflict of interest.

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
