## [Reviewer comments · Life Science Alliance]

Life Science Alliance

Plasticity in salt-bridge allows fusion-competent ubiquitylation of mitofusins and Cdc48 recognition

Vincent Anton, Ira Bunttenbroich, Ramona Schuster, Felix Babatz, Tânia Simões, Selver Altin, Gaetano Calabrese, Jan Riemer, Astrid Schauss, and Mafalda Escobar-Henriques

DOI: <https://doi.org/10.26508/lsa.201900491>

Corresponding author(s): Mafalda Escobar-Henriques, Institute for Genetics, Cologne Excellence Cluster on Cellular Stress Responses in Aging-Associated Diseases (CECAD), Center for Molecular Medicine Cologne (CMMC), University of Cologne

Review Timeline:

Submission Date:	2019-07-18
Editorial Decision:	2019-08-20
Revision Received:	2019-10-19
Editorial Decision:	2019-10-31
Revision Received:	2019-11-06
Accepted:	2019-11-07

Scientific Editor: Andrea Leibfried

Transaction Report:

August 20, 2019

Re: Life Science Alliance manuscript #LSA-2019-00491-T

Dr. Mafalda Escobar-Henriques
Institute for Genetics, Cologne Excellence Cluster on Cellular Stress Responses in Aging-Associated Diseases (CECAD), Center for Molecular Medicine Cologne (CMMC), University of Cologne
Joseph-Stelzmann-Straße 26
Cologne 50931
Germany

Dear Dr. Escobar-Henriques,

Thank you for submitting your manuscript entitled "Plasticity in salt-bridge allows fusion-competent ubiquitylation of mitofusins and Cdc48 recognition" to Life Science Alliance. For consistency, I asked the same set of reviewers to evaluate both studies on Fzo1 you submitted to us. Their reports are appended below.

As you will see, the reviewers appreciate your analyses of the salt bridge during the process of tethering, membrane fusion and disassembly and the role of ubiquitination in the process. We would thus like to invite you to submit a revised version of your manuscript to us, addressing the concerns raised by the reviewers. Importantly, some of your conclusions need to get further supported and further controls should get added. A few clarifications are also needed and the reviewers provide constructive input on how to restructure your work slightly. Addressing the reviewer concerns seems straightforward, but please do get in touch in case you would like to discuss individual points further. Note that addressing point 2d of reviewer #1 is not mandatorily needed for acceptance here.

Thank you for this interesting contribution to Life Science Alliance. We are looking forward to receiving your revised manuscript.

Sincerely,

B. MANUSCRIPT ORGANIZATION AND FORMATTING:

Reviewer #1 (Comments to the Authors (Required)):

Summary: In this manuscript, Anton et al., address the role of the lysine at position 464 in Fzo1-mediated fusion. Previous analysis of Fzo1-K464R indicated that the protein was not ubiquitinated when expressed in cells lacking *fzo1*, but that ubiquitination was reestablished if expressed with wild type Fzo1. Data presented in this manuscript indicate that ubiquitinated K464R does not support fusion. Therefore, authors seek to identify the molecular defect that renders K464R unable to fuse. Structural models based on BDLP and Mfn1 MGD suggested that K464 is part of a salt bridge network that includes R182 and D335. In these models, the D335 side chain forms a salt bridge with either R182 or K464, depending on the conformation of Fzo1. To test this prediction, double and triple charge swap mutant proteins were analyzed. In support of their model, only the triple charge swap mutant restored any Fzo1-mediated fusion activity in cells. To correlate each salt bridge with a distinct step in Fzo1-mediated fusion, authors present data that suggest that one salt bridge is required for generating tethered mitochondria while the second is not. Together, these data are consistent with a model where the primary defect in K464R is due to loss of a salt bridge, rather than lack of ubiquitination.

General comments: The manuscript presents and tests a model for Fzo1 conformational changes and data is consistent with the importance of movement between the GTPase domain and HB1 in Fzo1-mediated fusion. However, the conclusions would be significantly strengthened by the addition of some controls, outlined below. The experiments assessing the role of Cdc48 in Fzo1-mediated fusion are poorly justified in the current version of the manuscript.

Figure 1: To establish the premise for this study, Figure 1 shows concurrent loss of ubiquitination and fusion activity for Fzo1-K464R, Fzo1-T221A and Fzo1-T221A;K464R. Co-expressing with wild type Fzo1 or co-expressing the two mutant versions can reestablish ubiquitination of Fzo1-K464R and Fzo1-T221A. However, fusion activity was not restored when Fzo1-K464R and Fzo1-T221A are co-expressed in the same cell. Furthermore, these mutant forms do not fuse with a wild type Fzo1 on the opposite membrane in yeast mating assays for mitochondrial fusion activity. From this, authors conclude that ubiquitination is required, but not sufficient for fusion, and that K464R has a fusion defect beyond ubiquitination.

1a. It has been previously published (Griffin and Chan, 2006) that Fzo1-T221A does support fusion when co-expressed with Fzo1-V172P, Fzo1-L501A or Fzo1-L819P. It would be useful to include these controls to determine if Fzo1-K464R can be complemented in some cases. Given that mitochondria with Fzo1-K464R do not have a tethering defect as illustrated in the next figure, one might predict that it is not a null allele.

Figure 2: In structural models of Fzo1, K464 is predicted to be within a hinge region that is critical for Mfn1 function and also contains several amino acids that are associated with disease when altered in Mfn2. In particular, it has been shown that a series of salt bridges and hydrophobic interactions play an important role in a large conformational change between the GTPase domain and helical bundle in Mfn1. To determine if this region is also functionally important in Fzo1, D335 is identified as a potential salt bridge partner with K464 in the open conformation. Interestingly, R182 is structurally positioned to form a salt bridge with D335 in the closed conformation.

2a. It would be very helpful if the equivalent positions in Mfn1 and Mfn2 were given for each Fzo1 residue of interest discussed in this section as you are drawing information from Mfn1 structure and Mfn2 disease-associated alleles. An alignment could be included in the supplement, perhaps with a table to easily identify the homologous positions in each protein.

Consistent with these predictions, charge reversal of either D335 or R182 results in a non-functional Fzo1 protein that does not support fusion in cells. To determine if one salt bridge is sufficient for function, proteins with double charge reversals were tested for fusion activity. Neither Mfn1-D335R;R182D nor Mfn1-D335K;K464D supported fusion in cells lacking fzo1.

If each salt bridge occurs in a distinct conformational state of Fzo1, preventing the formation of a single salt bridge might trap Fzo1 in a discrete fusion state. To test this, isolated mitochondria were subject to docking conditions and tethered mitochondria were scored. While mitochondria with Fzo1-K464D were indistinguishable from wild type, mitochondria with Fzo1-R182E had a defect, suggesting that the closed conformation must be stabilized to tether mitochondria.

2b. Why is this assay performed in a ugo1-2 background?

2c. Why is a glutamic acid substitution tested here rather than the aspartic acid that was utilized in panel B of the same figure? Does R182D behave in the same way?

2d. Although this assay was previously published, it was by established by another group. Therefore, additional controls are required in this manuscript to validate the interpretation. This could include different nucleotide states, such as the presence of a transition state mimic to act as a positive control and trypsinized mitochondria to act as a negative control.

Figure 3: To test the model that the salt bridge is dynamic and that the side chain interacting with D335 depends on the conformational state of Fzo1, a triple substitution Fzo1 was generated: Fzo1-R182E; D335K; K464D. This version of Fzo1 restored some fusion activity in cells lacking fzo1. This was concomitant with partial restoration of the ubiquitination pattern. Yeast mating mitochondrial fusion assay confirmed fusion activity and indicates that fusion mediated by Fzo1-R182E; D335K; K464D is slower than wild type.

To emphasize the importance of the charges and therefore the proposed role in salt bridge formation, another version was tested where each amino acid was substituted with alanine. This version did not support fusion.

Figure 4: Authors test their prediction that Cdc48 will only recognize fusion-competent forms of Fzo1.

4a. The transition from Figure 3 to Figure 4 is not clear to this reviewer. Given that the previous data establish a role for K464 in conformational changes and the salt bridge formation, why revisit ubiquitination?

4b. Why are 4A and 4B separate panels? Why is Fzo1-K464R not tested? Based on Figure 1B, the double mutant version does not have the same abundance of ubiquitinated species as Fzo1 with a single amino acid substitution, yet all are equally non-functional. To separate ubiquitination from fusion, it would be preferable to test versions of Fzo1 that are most similar to wild type, in terms of ubiquitination.

4c. In addition to quantification of Fzo1 steady state levels, the interaction of Fzo1 with Cdc48 should be tested by co-immunoprecipitation to support the conclusion that Cdc48 only interacts with fusion competent Fzo1.

Authors propose that Cdc48 functions as a segregase in the mitochondrial fusion pathway, and

that impaired Cdc48 function would lead to accumulation of Fzo1 at fusion sites. To test this, a new assay is employed and Fzo1-eGFP distribution is examined in the presence and absence of Cdc48 function. In cells with wild type Cdc48, Fzo1 is distributed across the mitochondrial outer membrane and relatively few foci are observed. In contrast, cells with the *cdc48-2* allele have more Fzo1-eGFP foci.

4d. Additional controls are required to justify the interpretation of these experiments. Additional functional variants of Fzo1 should be assessed. For example, if Fzo1-eGFP only forms foci following fusion, fusion incompetent versions such as Fzo1-T221A and Fzo1-K464R should never form foci, even in the *cdc48-2* background.

Reviewer #2 (Comments to the Authors (Required)):

Overall impression:

Anton et al. examine the contributions of a dynamic trilateral salt bridge in mitochondrial membrane fusion as well as the role of CDC48 recognizing Fzo1 in a ubiquitin dependent manner. Overall the paper is well organized with very clean data and well described experiments. One punch line of the paper suggests that D335 switches interactions between R182 and K464 before and after GTP hydrolysis, respectively. The authors very convincingly show the importance of these interactions by generating complementary charge mutations in the three residues that can rescue defects seen when only one or two of these residues are mutated. Secondly, they show CDC48 can only recognize fusion competent ubiquitinated Fzo1 and is important for Fzo1 disassembly and recycling. This analysis in yeast provides strong evidence for a more detailed stepwise model of Fzo1 driven mitochondrial fusion, which they propose in Figure 5.

I recommend this manuscript be accepted for publication with minor revisions.

Summary:

The disease relevant K464 was previously shown to be required for the unique ubiquitination pattern of Fzo1, even though itself is not ubiquitinated. Based on recent crystal structures of the human homolog Mfn1 it is appreciated that K464 is located at the hinge region of Fzo1 and makes predicted electrostatic interactions with D335 only after GTP hydrolysis. Prior to GTP hydrolysis the elongated GTPase dimer shows D335 instead interacts with R182. The authors set out to further examine the importance of this dynamic trilateral interaction and how it alters downstream events in yeast mitochondrial membrane fusion.

In Figure 1/1S the authors establish that ubiquitination on K464R can occur when in the presence of a WT copy of Fzo1. Furthermore, ubiquitination can occur when a GTP hydrolysis mutant (T221A) is supplied as the alternative copy of Fzo1. When the two mutants are expressed together, they fail to tubulate mitochondria, despite being ubiquitinated. In the mating fusion assay, neither of these mutations correctly fuse mitochondria with a WT Fzo1 strain, suggesting that ubiquitination while required is not sufficient for fusion.

To further investigate the role of K464 the authors turned to the crystal structures of Mfn1 (human homolog), which show K464 exists in the hinge region of the GTPase and is proposed to form salt-bridge interactions with R182, E333 and D335. Of these three residues, both R182 and D335 show identical phenotypes to the K464R mutation. The authors nicely show that even the smallest side chain mutations (K to R and R to K) could not rescue the fusion defects. Different nucleotide bound states of the GTPase Mfn1 suggested the D335 residue could form alternating interactions with R182 and K464 depending on if the GTPase was GTP or GDP loaded, respectively. Pairwise complementary mutations were insufficient for rescuing ubiquitination or fusion, however, mutating all three residues to opposite charge partially restored Fzo1 ubiquitination and mitochondrial

membrane fusion.

Lastly, the authors try to determine why some Fzo1 mutants capable of ubiquitination cannot fuse mitochondria. To this end they show the K464, T221 double mutant is not being regulated by CDC48 activity like the WT. They also nicely show that the triple mutant (R182E, K464D, D335K) is targeted by CDC48 similar to the WT protein, highlighting the necessity of the hinge region and GTP cycling. Together with aggregation data in a DNM1-111 cdc48-2 mutant they show Fzo1 aggregates when CDC48 is inactivated.

Based on data in this paper and previous literature, the authors propose a more detailed model for the mechanisms driving Fzo1 activity.

Issues to resolve:

The troubleshooting of the mating fusion assay and switching to the galactose repression-based assay seems unnecessary and distracting to the reader. I would suggest this be removed from the text and figures.

A diagram of the hinge region with the CMT2A disease causing mutations and the yeast equivalents would be beneficial and could maybe take the place of Supplemental Figure 1E? Consider swapping the figure subpanels Fig. 1B and Supp. Fig. 1G as the data in 1G has one of the major conclusions and really nicely shows the ubiquitination occurring in the double mutant expressing strain.

For the mitochondria membrane tethering versus docking experiment, the number of counted events is different between the methods and the figure legend. Please address. Secondly, it is unclear how the events were binned between tethered and docked; was this done in a quantitative manner?

For consistency, please include a loading control from the right panel of Supplemental Figure 2E.

Reviewer #3 (Comments to the Authors (Required)):

This manuscript reports studies to uncover mechanistic insights into Fzo1 mediated mitochondrial fusion. The authors use mutagenesis and yeast studies, coupled with structural modelling approaches, to uncover that residues in a putative hinge-point region undergo salt bridge swapping during the process of tethering, membrane fusion and disassembly. They also suggest that Fzo1 ubiquitylation occur subsequent to tethering with disassembly afforded by Cdc48. They present a model that encompasses this work in the context of previous studies. To me, this is a relatively clear finding and worthy of publication. I have a few minor comments:

1. I felt that the authors have presented work that give very straight forward conclusions, yet beef it up with some experimental detail that seems redundant. In particular Fig S1E is not required nor the narrative about shutting down Fzo1 expression shown in Fig. 1C,D. This is an obvious experimental approach that one would expect should be employed in fusion assays.

2. The loading in 4B is poor with the salt bridge mutant expression in the cdc48-2 mutant. The authors should show a different experiment with equal loading and showing the Ub pattern changes more clearly.

3. In Fig. 4C, the authors note that they found an increase of Fzo1-GFP foci in cdc48-2 mutant cells, when compared to WT cells. However, it is unclear what these foci represent (stalled fusion assemblies?) and more importantly, why they only see 1-2 foci in the cells shown in the figure (arrow) yet quantify about 50/cell. Please clarify and show clearer images of the multiple foci.

4. There are some typos throughout - please proof read carefully.

Reply to Reviewer 1:

We are grateful that this reviewer appreciates our approach to test a model of Fzo1 conformational changes and consistencies with previous reports. We appreciate the importance of the suggested improvements and controls, which we fully addressed below:

Specific points:

1a "It has been previously published (Griffin and Chan, 2006) that Fzo1-T221A does support fusion when co-expressed with Fzo1-V172P, Fzo1-L501A or Fzo1-L819P. It would be useful to include these controls to determine if Fzo1- K464R can be complemented in some cases. Given that mitochondria with Fzo1-K464R do not have a tethering defect as illustrated in the next figure, one might predict that it is not a null allele."

These experiments were performed as suggested, but mitochondrial morphology was not rescued by co-expressing Fzo1^{K464R} together with Fzo1^{V172P} or Fzo1^{L819P} (see attached Figure). However, unlike Griffin and Chan, we could not observe the expected rescue of mitochondrial morphology when co-expressing Fzo1^{T221A} with Fzo1^{V172P} or Fzo1^{L819P}. Therefore, this experiment is inconclusive, so we did not include it in the manuscript.

1. 2a "It would be very helpful if the equivalent positions in Mfn1 and Mfn2 were given for each Fzo1 residue of interest discussed in this section as you are drawing information from Mfn1 structure and Mfn2 disease-associated alleles. An alignment could be included in the supplement, perhaps with a table to easily identify the homologous positions in each protein."

We followed this suggestion and now added the alignment with the equivalent positions in MFN1 and MFN2, the corresponding disease mutations and also a picture of the positions in the crystal model (Fig. S1 B).

2. 2b "Why is this assay performed in a ugo1-2 background?"

We apologize that this has been unclear in the previous version of the manuscript. The *ugo1-2* mutant was previously shown to accumulate docked mitochondria (Hoppins et al. 2009). As we could show no clear difference in docking between the two tested salt bridge mutants in the Δ *fzo1* mutant background, we speculated that this was due to clearance of the tethering complexes downstream of Ugo1 activity. Therefore, we performed this experiment in the *ugo1-2* background to stall the fusion process at the docked stage, allowing to test whether the Fzo1 mutants could reach this stage. This has been clarified in the manuscript (line 200- 204).

3. 2c "Why is a glutamic acid substitution tested here rather than the aspartic acid that was utilized in panel B of the same figure? Does R182D behave in the same way?"

Fzo1^{R182D} or Fzo1^{R182E} have been used in the different experiments for setup reasons: in Fig 2 B, Fzo1^{R182D} was chosen as it represents the direct swap with residue D335. In Fig 2 D, Fzo1^{R182E} was chosen as it represents the single mutant corresponding to the triple salt bridge swap mutant used in Fig 3, assuming that the two mutations of residue R182 should behave the same way. However, we share the concern of the reviewer and performed the tethering assay with Fzo1^{R182D}, which allowed to confirm our assumption. This control has been included in the manuscript (Fig 2 D).

4. 2d "Although this assay was previously published, it was by established by another group. Therefore, additional controls are required in this manuscript to validate the interpretation. This could include different nucleotide states, such as the presence of a transition state mimic to act as a positive control and trypsinized mitochondria to act as a negative control."

We are thankful for the suggestion of this very valid control. We now performed the tethering assay either in the presence of GTP γ S to determine the maximum tethering capacity or after

treatment of mitochondria with trypsin as a negative control. These controls behaved as expected and the corresponding figure and text have been included in the manuscript (Fig 2 D, lines 205- 207).

5. 4a “The transition from Figure 3 to Figure 4 is not clear to this reviewer. Given that the previous data establish a role for K464 in conformational changes and the salt bridge formation, why revisit ubiquitination?”

We acknowledge that the transition from Fig 3 to Fig 4 might have been unclear, and have now altered the text. In sum, we wanted to unify two main findings –ubiquitylated and either fusion competent or fusion incompetent Fzo1 mutants, from the perspective of how conformational rearrangements affect a chaperone dedicated to recognize ubiquitin on its substrates. Indeed, in Fig 1 we could show that Fzo1^{K464R} shows no fusion activity, even though it can be ubiquitylated. Nevertheless, Fig 3 shows we could rescue fusion activity in the K464 mutant, by further mutation of the two other salt bridge residues R182 and D335. This is likely due to the rescue of conformational changes, which cannot occur on Fzo1^{K464R} even when ubiquitylation is rescued. Thus, these tools were used to understand the requirements of Fzo1 for Cdc48 recognition. The manuscript has been adjusted to make this reasoning more obvious (lines 236- 241).

6. 4b “Why are 4A and 4B separate panels? Why is Fzo1-K464R not tested? Based on Figure 1B, the double mutant version does not have the same abundance of ubiquitinated species as Fzo1 with a single amino acid substitution, yet all are equally non-functional. To separate ubiquitination from fusion, it would be preferable to test versions of Fzo1 that are most similar to wild type, in terms of ubiquitination.”

We apologize that the difference between Fig 4 A and 4 B might have been unclear. In sum, 4 A analyses fusion-incompetent and 4 B fusion-competent forms of ubiquitylated Fzo1. Further, we have now included the K464R and T221A mutant variants and improved labelling.

7. 4c “In addition to quantification of Fzo1 steady state levels, the interaction of Fzo1 with Cdc48 should be tested by co-immunoprecipitation to support the conclusion that Cdc48 only interacts with fusion competent Fzo1.”

This valid suggestion has been addressed (Fig 4 C), confirming Cdc48 interaction with fusion competent Fzo1, wild type or Fzo1^{R182E;D335K;K464D}, but not with fusion incompetent Fzo1^{K464R}.

8. 4d “Additional controls are required to justify the interpretation of these experiments. Additional functional variants of Fzo1 should be assessed. For example, if Fzo1-eGFP only forms foci following fusion, fusion incompetent versions such as Fzo1-T221A and Fzo1-K464R should never form foci, even in the cdc48-2 background.”

We agree with the reviewer that additional controls are required for the interpretation of these results especially because it was shown that Fzo1 foci form as soon as *trans*- oligomerization occurs (Brandt et al., 2016). As suggested, we now analysed Fzo1^{T221A}-GFP and Fzo1^{K464R}-GFP in both wt and *cdc48-2* background. Indeed, both Fzo1^{T221A}-GFP and Fzo1^{K464R}-GFP formed foci even in presence of wild type Cdc48. This Figure has been included (Fig 4 D) and the text has been modified accordingly (lines 258- 261).

Reply to Reviewer 2:

We are thankful that this reviewer appreciates the organization and quality of the data presented. Further we acknowledge the reviewer's appreciation for the findings on the dynamic trilateral interactions and downstream consequences for ubiquitylation and Cdc48- mediated clearance of Fzo1 oligomers. Specific concerns have been addressed below.

Specific points:

1. *"The troubleshooting of the mating fusion assay and switching to the galactose repression-based assay seems unnecessary and distracting to the reader. I would suggest this be removed from the text and figures."*

The corresponding figure and text have now been removed from the manuscript.

2. *"A diagram of the hinge region with the CMT2A disease causing mutations and the yeast equivalents would be beneficial and could maybe take the place of Supplemental Figure 1E?"*

The suggested figure has been added to the manuscript, corresponding to Fig S1 B.

3. *"Consider swapping the figure subpanels Fig. 1B and Supp. Fig. 1G as the data in 1G has one of the major conclusions and really nicely shows the ubiquitination occurring in the double mutant expressing strain."*

We are thankful for the very valid suggestion and changed the order of the figures accordingly, which now consist of Fig 1 C and S1 D.

4. *"For the mitochondria membrane tethering versus docking experiment, the number of counted events is different between the methods and the figure legend. Please address."*

We apologize for this oversight and corrected it (lines 707- 709 and 725- 726).

5. *"Secondly, it is unclear how the events were binned between tethered and docked; was this done in a quantitative manner?"*

We recognize that it might have been unclear how tethering and docking events were defined. A description of how these events were binned has now been added to the methods section (lines 445- 452). In brief, mitochondria were quantified as tethered when the contact between mitochondria could be identified by distinct membrane contact and changes of membrane curvature. Mitochondria were quantified as docked when this contact site further extended to over at least one third of the diameter of the mitochondria, accompanied by deformation of the usual membrane curvature, recognized by the presence of two parallel membranes of the apposing mitochondria.

6. *"For consistency, please include a loading control from the right panel of Supplemental Figure 2E."*

The loading control for this experiment has been added.

Reply to Reviewer 3:

We are very glad that this Reviewer appreciates the presented model and the clarity of the findings. All comments have been implemented and are listed below.

Specific points:

1. *“I felt that the authors have presented work that give very straight forward conclusions, yet beef it up with some experimental detail that seems redundant. In particular Fig S1E is not required nor the narrative about shutting down Fzo1 expression shown in Fig. 1C,D. This is an obvious experimental approach that one would expect should be employed in fusion assays.”*

The corresponding Figures and text have been removed from the manuscript.

2. *“The loading in 4B is poor with the salt bridge mutant expression in the cdc48-2 mutant. The authors should show a different experiment with equal loading and showing the Ub pattern changes more clearly.”*

The corresponding experiment has been replaced.

3. *“In Fig. 4C, the authors note that they found an increase of Fzo1-GFP foci in cdc48-2 mutant cells, when compared to WT cells. However, it is unclear what these foci represent (stalled fusion assemblies?) and more importantly, why they only see 1-2 foci in the cells shown in the figure (arrow) yet quantify about 50/cell. Please clarify and show clearer images of the multiple foci.”*

We apologize that this figure was not well explained. Now, the method of quantification was further detailed and adjusted (Fig 4 D). Previously, we had total foci numbers counted in 100 cells. In addition, this experiment has been replaced, as nicely suggested by reviewer 1. Now, we could show a strong accumulation of foci when expressing Fzo1^{T221A}-GFP or Fzo1^{K464R}-GFP even in the wt Cdc48 (Fig 4 D). As Fzo1 foci form as soon as *trans*- oligomerization occurs (Brandt et al., 2016), these foci therefore indeed likely represent stalled fusion complexes.

4. *“There are some typos throughout - please proof read carefully.”*

The manuscript has been proof read and typos have been eliminated.

October 31, 2019

RE: Life Science Alliance Manuscript #LSA-2019-00491-TR

Dr. Mafalda Escobar-Henriques
Institute for Genetics, Cologne Excellence Cluster on Cellular Stress Responses in Aging-Associated Diseases (CECAD), Center for Molecular Medicine Cologne (CMMC), University of Cologne
Joseph-Stelzmann-Straße 26
Cologne 50931
Germany

Dear Dr. Escobar-Henriques,

Thank you for submitting your revised manuscript entitled "Plasticity in salt-bridge allows fusion-competent ubiquitylation of mitofusins and Cdc48 recognition". As you will see, the reviewers appreciate the changes introduced in revision, and we would thus be happy to publish your paper in Life Science Alliance, pending final revisions necessary to meet our formatting guidelines:

- please add a callout in the manuscript text to figures S4B
- please make sure that the bar graphs are sufficiently explained in all figure legends - it is not always clear what the triangles, circles, rectangles mean when looking at individual figure legends
- please add scale bars to the images where missing

A. FINAL FILES:

- An editable version of the final text (.DOC or .DOCX) is needed for copyediting (no PDFs).
- High-resolution figure, supplementary figure and video files uploaded as individual files: See our detailed guidelines for preparing your production-ready images, <http://www.life-science-alliance.org/authors>
- Summary blurb (enter in submission system): A short text summarizing in a single sentence the

study (max. 200 characters including spaces). This text is used in conjunction with the titles of papers, hence should be informative and complementary to the title. It should describe the context and significance of the findings for a general readership; it should be written in the present tense and refer to the work in the third person. Author names should not be mentioned.

B. MANUSCRIPT ORGANIZATION AND FORMATTING:

Sincerely,

Reviewer #1 (Comments to the Authors (Required)):

The authors have addressed all questions and concerns.

Reviewer #2 (Comments to the Authors (Required)):

The authors did a great job addressing my concerns and appear to have done so for the other reviewers as well. I recommend that this manuscript be accepted for publication.

Reviewer #3 (Comments to the Authors (Required)):

I am happy with the revisions made - the authors have done an excellent job addressing all comments and I feel that the manuscript is suitable for publication. The figures are really lovely too.

November 7, 2019

RE: Life Science Alliance Manuscript #LSA-2019-00491-TRR

Dr. Mafalda Escobar-Henriques
Institute for Genetics, Cologne Excellence Cluster on Cellular Stress Responses in Aging-Associated Diseases (CECAD), Center for Molecular Medicine Cologne (CMMC), University of Cologne
Joseph-Stelzmann-Straße 26
Cologne 50931
Germany

Dear Dr. Escobar-Henriques,

Thank you for submitting your Research Article entitled "Plasticity in salt-bridge allows fusion-competent ubiquitylation of mitofusins and Cdc48 recognition". It is a pleasure to let you know that your manuscript is now accepted for publication in Life Science Alliance. Congratulations on this interesting work.

DISTRIBUTION OF MATERIALS:

Again, congratulations on a very nice paper. I hope you found the review process to be constructive and are pleased with how the manuscript was handled editorially. We look forward to future exciting

submissions from your lab.

Sincerely,
